# Structural and social factors affecting COVID-19 vaccine uptake among healthcare workers and older people in Uganda: A qualitative analysis

**Sande Slivesteri**[1]*, **Agnes Ssali**[1,2], **Ubaldo M. Bahemuka**[1], **Denis Nsubuga**[1], **Moses Muwanga**[3], **Chris Nsereko**[3], **Edward Ssemwanga**[4], **Asaba Robert**[5], **Janet Seeley**[1,2], **Alison Elliott**[1,6], **Eugene Ruzagira**[1,7]

1 MRC/UVRI & LSHTM Uganda Research Unit, Wakiso, Uganda, 2 Department of Global Health and Development, London School of Hygiene and Tropical Medicine, London, United Kingdom, 3 Entebbe Regional Referral Hospital, Wakiso, Uganda, 4 Villa Maria Hospital, Kalungu, Uganda, 5 Our Lady of Consolata, Kisubi Hospital, Wakiso, Uganda, 6 Department of Clinical Research, London School of Hygiene and Tropical Medicine, London, United Kingdom, 7 Department of Infectious Disease Epidemiology, London School of Hygiene and Tropical Medicine, London, United Kingdom

* Sandemakslivesters@gmail.com

**Data Availability Statement:** Due to the characteristics of the data and the consent given by participants, with a guarantee of anonymity, we are

## Abstract

The COVID-19 vaccine rollout programme in Uganda was launched in March 2021 for priority groups: Healthcare Workers (HCWs), older persons ($\geq$50 years), and persons with chronic conditions. Misinformation, distrust in healthcare systems, and cultural beliefs, pose significant challenges to vaccine uptake. We describe the social and structural factors affecting the uptake of COVID-19 vaccines among HCWs and older people in Uganda. Between September and October 2021, we conducted 33 in-depth interviews with 25 HCWs aged 21–63 years from three hospitals in central Uganda and eight older people from Wakiso district. Participant selection was purposive, based on sex, occupation, education, cadre of HCWs and vaccination status. We explored participants' knowledge, beliefs, personal experiences, barriers, and facilitators to vaccine uptake as well as suggestions for future COVID-19 vaccine rollout. Interviews were audio-recorded, transcribed and translated into English, coded, and analysed by theme. Twenty-two of the 25 (88%) HCWs and 3 of the 8 (38%) older people had received at least one dose of the COVID-19 vaccine at the time of interview. The structural facilitating factors to vaccine uptake included access to correct information, fear of a risky work environment, and mandatory vaccination requirements especially for frontline HCWs. Age, chronic health conditions, and the fear of death were facilitating factors for older people. Misconceptions about COVID-19 vaccines and fear of side effects were common social barriers for both groups. Long distances to vaccination centres, vaccine stock-outs, and long queues at the vaccination centres were specific barriers for older people. The prerequisite of signing a consent form was a specific structural barrier for HCWs. Future roll out of new vaccines should have a comprehensive information dissemination strategy about the vaccines. Improved access to vaccines through

unable to include all the original data. This precaution is necessary, as certain details could potentially compromise the anonymity of the participants. Hence, the data forming the basis of this paper has been incorporated into the Supplementary Information and is also accessible via: https://doi.org/10.17037/DATA.00003700, ensuring that any mention of names, age, and specific job identifiers has been excluded. Additional data will be provided upon request. All correspondence and applications related to this data will be handled by the MRC/UVRI and LSHTM Uganda Research Unit. Researchers may apply to access limited, anonymized information contained within the interview transcripts. Please consult the codebook and state the research topics/questions in which you are interested when applying for access. If there is a recognized risk of indirect identification from the provision of selected quotes, applicants (or their host institution) will be asked to sign a data transfer agreement, establishing confidentiality conditions. Requests should be submitted to LSHTM Data Compass via email: researchdatamanagement@lshtm.ac.uk.

**Funding:** This work was funded through the Makerere University-Uganda Virus Research Institute Centre of Excellence for Infection and Immunity Research and Training (MUII) to AE. MUII was supported through the DELTAS Africa Initiative (Grant no. DEL-15-004). The DELTAS Africa Initiative was an independent funding scheme of the African Academy of Sciences (AAS), Alliance for Accelerating Excellence in Science in Africa (AESA) and supported by the New Partnership for Africa's Development Planning and Coordinating Agency (NEPAD Agency) with funding from the Wellcome Trust and the UK Government. The work was conducted at the MRC/UVRI and LSHTM Uganda Research Unit which is jointly funded by the UK Medical Research Council (MRC) part of UK Research and Innovation (UKRI) and the UK Foreign, Commonwealth and Development Office (FCDO) under the MRC/FCDO Concordat agreement and is also part of the EDCTP2 programme supported by the European Union. Alison Elliott (AE) is supported in part by the NIHR (NIHR134531) using UK aid from the UK Government to support global health research. The views expressed in this publication are those of the author(s) and not necessarily those of the funders or the UK government. The funders had no role in study design, data collection and analysis, decision to publish, or preparation of the manuscript.

**Competing interests:** The authors have declared that no competing interests exist.

community outreach, reliable vaccine supplies and addressing vaccine misinformation, may enhance COVID-19 vaccine uptake.

## Background

By June 2023, the coronavirus (COVID-19) pandemic had caused 12.2 million confirmed cases with 256,542 deaths in Africa and 170,544 confirmed cases with 3,632 deaths in Uganda [1]. The pandemic placed a considerable strain on the public health system, notably in its initial phase in 2020. Besides lockdown prevention measures causing great mobility challenges in accessing health facilities, many public health facilities scaled down on antenatal, HIV, vaccination, and other health services to focus on COVID-19, especially in 2020–2021 [2–4].

Several vaccines have proved to be safe and effective against laboratory-confirmed SARS-CoV-2 virus infection and symptomatic or severe COVID-19 disease [5–7]. Since December 2020, several countries initiated COVID-19 vaccine rollout programmes, with 67% of the world population fully vaccinated and 31.5% vaccinated with at least one booster or additional dose by June 2023. Vaccination rollout has been reportedly slower in Africa, with only 49.7% of the African population fully vaccinated [1, 8]

In Uganda, COVID-19 vaccine rollout was launched on 10th March 2021 targeting at-risk groups: healthcare workers, security personnel, people aged 50 years and above, and those aged 18–50 years with comorbidities [9–11]. COVID-19 vaccination was scaled up to target all persons aged 18 years and above and children of 12 years above in 2022 [11, 12].

Findings from a study in South Africa indicated that acceptance of a new COVID-19 vaccine was influenced by different factors including age, employment status, urbanity, and geographical location [13]. Another study among high-risk populations in Uganda showed that 70% of the participants were open to receiving the COVID-19 vaccine with the probabilities being four times higher in men versus women [9].

However, other studies reported concerns about the vaccine's expedited development and approval process, potential side effects, and efficacy [14–17]. Such concerns contribute to vaccine hesitancy, which has been reported as a significant challenge to disease prevention and control worldwide. The WHO describes vaccine hesitancy as the delay in the acceptance or complete refusal of vaccines [18].

Structural, social, and contextual factors may contribute to vaccine confidence and affect the uptake of the COVID-19 vaccine among HCWs and older people in Uganda. We set out to describe the structural, social, and contextual factors that influence the uptake of COVID-19 vaccines among HCWs and older people aged $\geq$ 50 years using the Social Ecological Model (SEM) to structure our analysis.

## Theoretical framework

The Social Ecological Model (SEM) provides a framework for exploring the relationship between social and structural factors and the physical environment and how such factors influence and shape the decision-making process of individuals, including their health-seeking behaviours [19, 20]. The model has been used to describe individuals' behaviours using measurements comprising of intrapersonal (within the individual), interpersonal (between individuals), institutional, community and public policy factors to provide a framework for understanding the interplay between these levels and how they shape or influence an individual's actions and choices [20, 21].

Using this framework, we explored participants' knowledge, beliefs, and attitudes about COVID-19, personal experience about COVID-19 vaccines, facilitators and barriers to vaccine uptake, and opinions on the future of vaccine rollout **(See Fig 1 below).**

The intrapersonal level includes individual characteristics, such as age, education, income, and health history, the interpersonal level comprises the relationship and interaction with others in a person's closest social circle, such as friends, partners, and family members, all of whom influence a person's behaviours including health-seeking behaviours. The institutional layer includes the social organisation characteristics, rules, and regulations for operation. The

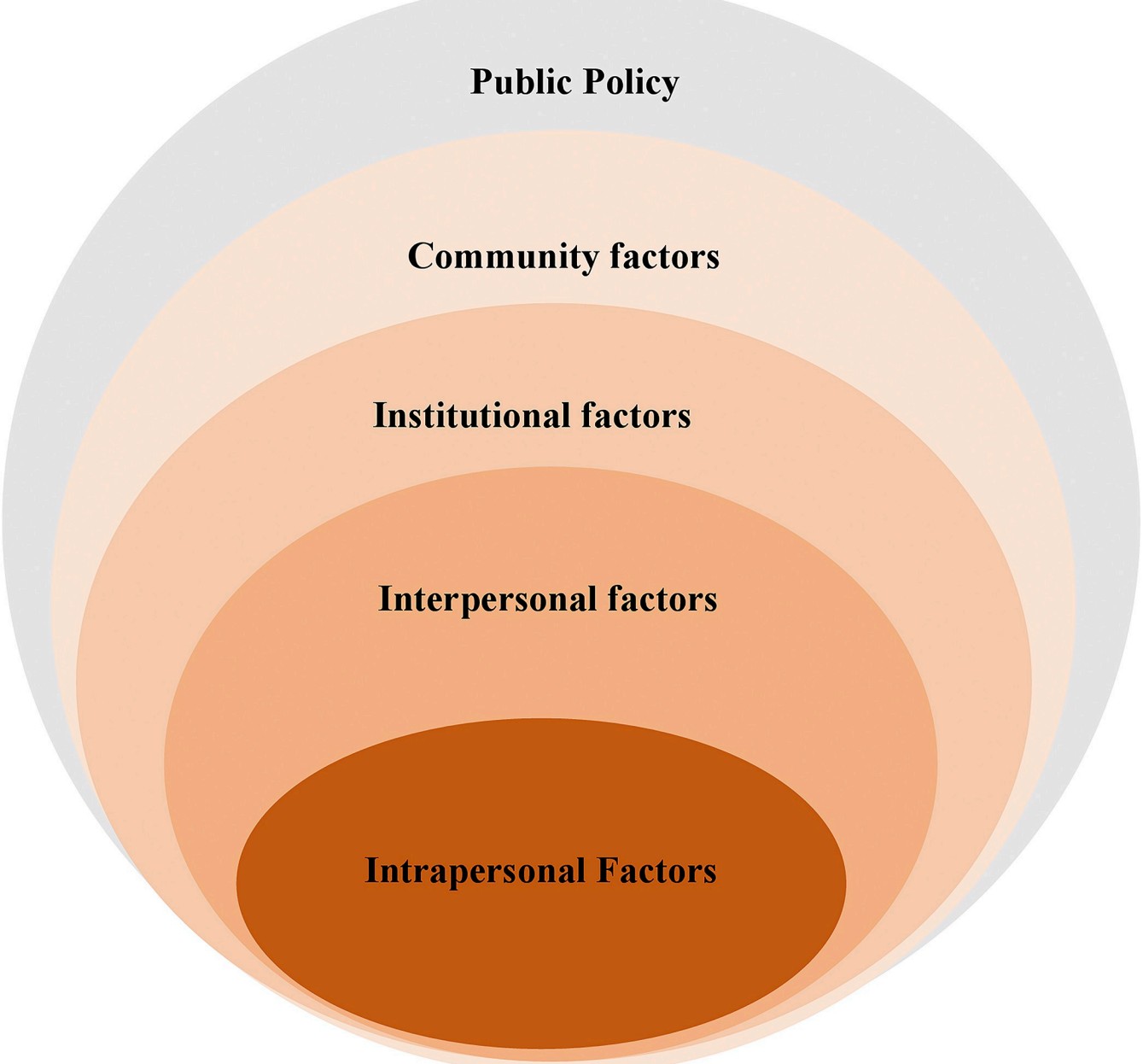

**Fig 1. The social ecology of health promotion interventions.** Adapted from McLeroy, K. R., Steckler, A. and Bibeau, D. (Eds.) (1988).

community layer covers the settings in which people have social relationships, such as schools, workplaces, and neighbourhoods, and the characteristics of these settings that affect health. Finally, the public policy layer includes the broad societal factors that favour or impair health including state regulations.

## Methods

### Study design and setting

This was a qualitative methods study conducted among a subset of participants enrolled between May and October 2021 in a prospective study to investigate acceptability and immunogenicity of COVID-19 vaccines among 597 HCWs and 150 older people ($\geq$50 years) in Uganda. The HCWs were enrolled from three hospitals: I) Entebbe regional referral public hospital, located in Entebbe municipality, 44km from Kampala, the capital city of Uganda; II) Our Lady of Consolata Kisubi Hospital, a Private Not-For-Profit (PNFP) hospital located approximately 28km from Kampala, and Villa Maria Hospital, a PNFP hospital in Kalungu district, approximately 138km from Kampala. The older people were participants in the Well-being of Older People Study (WOPs) that has been ongoing in the semi-urban Wakiso district and rural Kalungu district for 12 years. The objective of WOPs is to describe the roles, health problems (physical and mental) and social well-being of older people directly and indirectly affected by HIV/AIDS, with special attention to the effects of the introduction of Antiretroviral Therapy (ART). In the current study, we enrolled HCWs from the three hospitals and older people from Wakiso District.

### Study participants, and sampling

Using the participant identification logs of the COVID-19 vaccine acceptability and immunogenicity study, we conducted a purposive sampling process to identify and invite 33 individuals who had been enrolled in the preceding three weeks to participate in the in-depth interviews.

We aimed to capture a wide range of perspectives by purposively including participants with diverse demographic characteristics. Selection was based on sex, occupation, education, cadre of the HCWs (doctors, nurses, laboratory technicians, hospital support staff and administrators). We also ensured that our sample included participants with different vaccination status. This was important to explore the perceptions and experiences of those who had been vaccinated and those who had not.

To account for potential variations in vaccine acceptability among different healthcare settings, we selected participants from three distinct hospitals: Kisubi hospital (n = 8), Entebbe Regional Referral hospital (n = 8), and Villa Maria hospital (n = 9). Additionally, we included 8 older participants from WOPs. This approach allowed us to move beyond solely studying vaccine acceptability among HCWs and explore the perspectives of a more diverse sample within a COVID-19 at-risk population. The combination of these factors guided our selection process, ensuring that we had a diverse group of participants for the in-depth interviews.

### Data collection

Social science research assistants (RAs) contacted the selected individuals by telephone and requested their participation in the in-depth interviews. The RAs and the identified participants agreed on the date, time, and convenient venue to discuss study information, and if they consented, interviews were conducted on the same day.

Interviews were conducted using a semi-structured interview guide (see S1 Text) that included the following topics: knowledge, beliefs, and attitudes towards COVID-19 vaccines; enabling factors and barriers to uptake of COVID-19 vaccines, and personal views on the future of the COVID-19 vaccine rollout programme. The interviews were audio-recorded if the participant agreed, or notes were taken if the participant declined voice recording. Interviews were conducted in the local vernacular language (Luganda). A few interviews were conducted in English for the participants who preferred this. Each interview lasted about 45 minutes. A senior social scientist conducted regular debriefing meetings with the RAs to review the completeness of the data and identify areas to improve on in subsequent interviews.

## Data management and analysis

Anonymized audio files were transferred onto encrypted password-protected computers, transcriptions and translations were done by the RAs, and anonymized transcripts were transferred to a secure data server at the Medical Research Council /Uganda Virus Research Institute and London School of Hygiene and Tropical Medicine (MRC/UVRI and LSHTM) Uganda Research Unit.

The data analysis process involved an iterative approach with two research assistants, under the guidance of a senior social scientist. The research team collaboratively reviewed and discussed transcripts to identify and reach a consensus on common codes. A codebook was developed based on a priori data categorization and emergent categories that came from the coding process. The codebook included definitions and examples for each code to facilitate consistent coding (see S2 Text and S3 Text).

To ensure consistency and reliability of the coding process, two research assistants independently coded a subset of transcripts, compared results and any differences in coding were resolved through discussion led by the senior social scientist until a consensus was reached. Patterns from the data led to the themes identified in this paper which are informed by the layers from the SEM.

## Ethical considerations

The study was approved by the Uganda Virus Research Institute Research Ethics Committee (UVRI REC GC/127/21/03/813), the Uganda National Council for Science and Technology (UNCST SS767ES, 27-04-2021), and the London School of Hygiene and Tropical Medicine Research Ethics Committee (25997). We obtained administrative clearance from all the collaborating hospitals to conduct the study. Written informed consent to participate in the interviews was obtained at enrolment in the main study of acceptability and immunogenicity of COVID-19 vaccines. Before each interview, RAs verbally checked to confirm that participants were still interested in taking part in the in-depth interview. All interviews were conducted in a safe and private place to ensure participants' privacy and confidentiality.

## Results

### Demographic characteristics of study participants

Among the eight older people, the mean age was 68.8 (SD 6.7) years, 50% were female, 75% were Christian, 50% had attained at least secondary education level and 37.5% had received at least one dose of the vaccine (Table 1). Among the 25 HCWs, the mean age was 37.8 (SD 11.2) years, 32% were female, all were Christian, 68% had at least a diploma/bachelor's degree or other higher-level education, and 92% had received at least one dose of SARS-CoV2 vaccine (Table 1).

**Table 1. Characteristics of study participants.**

| | Older persons (N = 08) | Healthcare workers (N = 25) |
|---|---|---|
| **Characteristics** | **n (%)** | **n (%)** |
| **Sex** | | |
| Female | 4 (50) | 8 (32) |
| Male | 4 (50) | 17 (68) |
| **Mean age (SD)** | 67.8 (6.7) years | 37.8 (11.02) years |
| **Age group (years)** | | |
| 20–30 | - | 8 (32) |
| 31–40 | - | 8 (32) |
| 41–50 | - | 6 (24) |
| 51–60 | 1 (12.5) | 2 (8) |
| 61–70 | 4 (50) | 1 (4) |
| 71–80 | 3 (37.5) | - |
| **Religion** | | |
| Christian | 6 (75) | 25 (100) |
| Muslim | 2 (25) | - |
| **Highest Education Levels** | | |
| Incomplete primary | 2 (25) | - |
| Complete primary | 2 (25) | - |
| Incomplete O' level | - | 1 (4) |
| Complete O' level | 2 (25) | 4 (16) |
| Post O' level certificate and Vocational | 2 (25) | 2 (08) |
| Diploma | - | 4 (16) |
| Degree and above | - | 14 (56) |
| **Occupation** | | |
| Market vendors | 2 (25) | |
| Small business owners | 2 (25) | |
| Farmer | 1 (12.5) | |
| Teacher | 1 (12.5) | |
| Mechanic | 1 (12.5) | |
| Volunteer counsellor | 1 (12.5) | |
| Doctors | - | 3 (12) |
| Nurses | - | 6 (24) |
| Laboratory technicians | - | 3 (12) |
| Hospital Administrators | - | 2 (16) |
| Human resources | - | 2 (8) |
| Physiotherapist | - | 1 (4) |
| Radiographer | - | 1 (4) |
| Health economist | - | 1 (4) |
| Security guards | - | 3 (12) |
| Cleaners | - | 2 (8) |
| Records manager | - | 1 (4) |
| **District (setting)** | | |
| Kalungu (rural) | - | 9 (36) |
| Wakiso (semi-urban/urban) | 8 (100) | 16 (64) |
| **Hospitals** | | |
| Villa Maria Hospital | - | 9 (36) |
| Entebbe Regional Referral Hospital | - | 8 (32) |

(*Continued*)

**Table 1.** (Continued)

|  | Older persons (N = 08) | Healthcare workers (N = 25) |
|---|---|---|
| Our Lady of Consolata, Kisubi Hospital | - | 8 (32) |
| **Vaccination status** |  |  |
| Vaccinated (at least one dose) | 3 (37.5) | 23 (92) |
| Not vaccinated | 5 (62.5) | 2 (8) |

N = number; **SD** = standard deviation

We present our results ordered by the structure of the SEM, summarised in **Fig 2** below.

## Barriers to vaccination

**Intrapersonal barriers.** We captured participants' barriers to vaccine uptake based on their individual attributes, knowledge, beliefs, and attitudes towards COVID-19 vaccines.

Having chronic conditions/illnesses such as HIV, diabetes, hypertension, and allergies was described as a barrier to vaccine uptake by HCWs and older people. Participants shared the belief that when a person with such chronic conditions have the COVID-19 vaccine, it worsens their already existing chronic health condition, even when people with chronic health conditions are prioritised for the COVID-19 vaccination.

"*I have a disease called allergy, that is the only reason, why I have not been vaccinated yet"* Female, 47 years, HCW, not vaccinated.

The older people reported that HCWs advised people with chronic diseases and conditions to delay receiving the vaccine.

"*Whenever I get there (to the vaccination centres), the health workers tell me that the blood pressure is high and urge that I should wait for it to reduce before I take the vaccine"* Female, 63 years, older person, not vaccinated.

## Interpersonal barriers

Individuals' interactions with people close to them or people in their families/communities shape their behaviours, attitudes, and preferences, and influence their decision-making including health-seeking behaviours. The barriers included myths and misconceptions and fear of the effect the vaccine might have on them.

The myths and misconceptions attributed to social media posts and what other people were saying within homes and communities were reported to have caused fears and worries among the HCWs and older people about the vaccine. The myths and misconceptions that vaccines were intended to kill Africans, cause infertility and the linking of the COVID-19 vaccines to acts of "devil/satanic worship" were barriers to vaccine uptake. There was a reported misconception that people who received the vaccine would die two years after vaccination. There were also concerns about infertility:

"*I am not vaccinated because I have had that belief, that COVID-19 vaccines cause infertility*" Female, 23 years, HCW, not vaccinated.

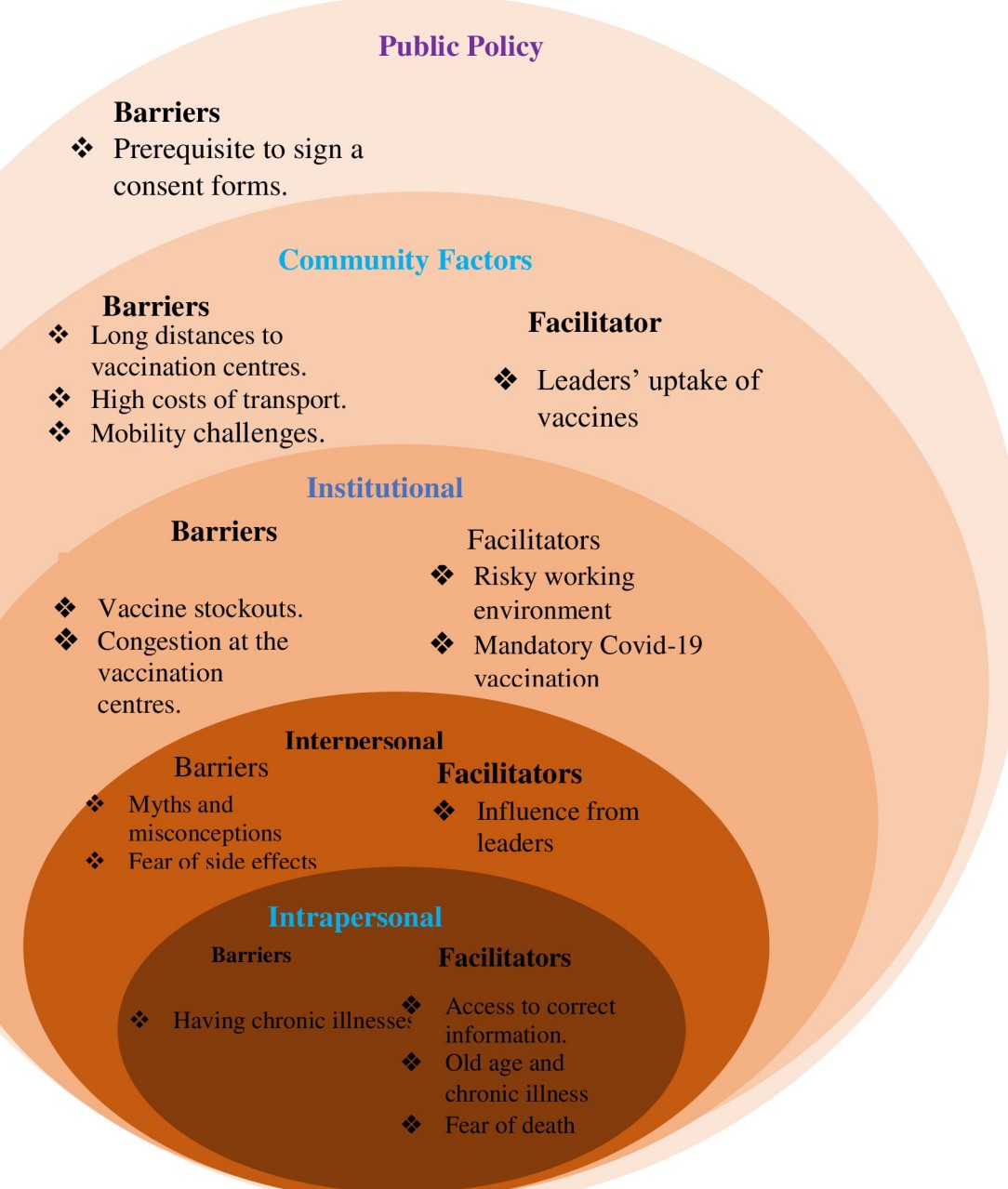

**Fig 2. Structural, social and contextual factors influencing COVID-19 vaccine uptake.**

"*They would say that in men it (vaccines) causes impotence and in female, you would be barren because it was a new drug, people did not know about it. So, those who had not yet had children said let me first wait and see*" Female, 44 years, HCW, vaccinated.

" *I don't want to be vaccinated because I heard about the infertility thing and the side effects like general body weakness and things of that kind*" Female, 23 years, HCW, not vaccinated.

The HCWs and older people described their fears of known and anticipated vaccine side effects as barriers to vaccine uptake. They attributed these vaccine fears to social media posts.

"*I usually see these videos and children show them to me like my daughter, they have phones, and they show them to me [asking] `have you seen?'. They have said this and that and the vaccine they have brought this time round is fake*" Female, 62 years, older person, vaccinated.

HCWs described the concerns about vaccine side effects shared by some people which had an impact on the vaccine uptake.

"*To the patients, when they come, and they do not have a fever, they happen to start feeling fever and severe headache after vaccination. They conclude that they have been given poison*" Female, 29 years, HCW, vaccinated.

## Institutional barriers

Vaccine stockouts and long wating times were reported to have negatively impacted on the vaccine uptake.

HCWs and older people reported that vaccine stock-outs at the vaccination centres were a barrier to vaccine uptake.

"*I went to Kisubi hospital to be vaccinated and there were no vaccines. They gave me a telephone number to call them and check if they have the vaccines. I have called them twice, and the vaccines are not yet available*" Male, 68 years, older person, not vaccinated.

Some HCWs reported sending away people who came to the health facilities for vaccination because of vaccine stock-outs. The failure to find vaccines at the vaccination centres after travelling long distances was a reported setback to vaccine uptake.

"*Limited vaccine doses [. . .] like now people intending to receive the vaccines come, and you tell them that the vaccines are not available, and people move several times to the hospital and do not find the vaccine*" Female, 29 years, HCW, vaccinated.

The congestion at vaccination centres, coupled with the lack of special arrangements to facilitate easy access to vaccines for older people, created barriers to vaccine uptake. The older people reported that they could not stand in long queues at the vaccination centres to receive the vaccines.

"*In the beginning, being a new thing (COVID-19 vaccine), I know that it attracts many people. In my life, I do not want to struggle and be in a long queue like many people standing in the same place. I would not go there (hospital) because people were many people at the vaccination sites*" Male, 68 years, older person, not vaccinated.

## Community barriers

Long distances to vaccination centres coupled with high transport costs and mobility challenges are community barriers reported.

The older people reported transport challenges getting to the vaccination centres within the communities as a barrier to vaccine uptake. The long distances to the vaccination centres and

the COVID-19-related public transportation restrictions (the requirement for public transport vehicles to operate at half capacity), led to increased transport costs.

"*I do not have transport to UVRI Entebbe or Kisubi hospital, that is the main reason why I am not vaccinated, nothing else*" Male, 76 years, older person, not vaccinated.

Participants emphasised that transport costs and age-related mobility challenges were a major hindrance to vaccine uptake.

"*The older person has difficulties taking her/him from where she/he is to the vaccination centres, they cannot move, many do not have money for transport,*" Male, 68 years older person, not vaccinated.

The long distance to vaccination centres was reported as a challenge by some HCWs and older people. They highlighted that COVID-19 vaccines were not available in lower-level health facilities, such as Health Centres II, which operate at the community level. HCWs emphasized that vaccines were only accessible at Health Centre III and hospital-level health facilities which were often in towns and larger trading centres. This was reported to have influenced vaccine access because people had to move long distances to the vaccination centres.

"*The distance to the vaccination centres is long.*" Female, 76 years, older person, not vaccinated.

"*Some people's homes are too far from the health centres that are giving the vaccines because there are villages that are far from the health facilities and vaccines are not available in those communities* Male, 38 years, HCW, vaccinated.

## Public policy barrier

There were particular factors associated with the roll-out of the COVID-19 vaccination campaign which differed from other vaccination programmes and raised concerns. Some HCWs complained about the unusual process of signing a consent form before being given the vaccine. This requirement caused fears and doubts about the safety of the COVID-19 vaccine.

"*Why do they first want us to consent? So, they mean that if I have any problem, there is nowhere I can report because I consented. So, most people fear that consent form*. Female, 29 years, HCW, vaccinated.

## Facilitators to vaccination

**Intrapersonal facilitators.** HCWs and older people reported having access to correct information on COVID-19 vaccines as a facilitator to vaccine uptake. The World Health Organization (WHO) COVID-19 updates, the Continuous Medical Education series (CMEs), the updates from the Ministry of Health, training by the District Health Teams (DHT), the internet, and social media were reported as specific sources of information for HCWs. The news media (television and radio) were the information sources for older people and some HCWs. Some HCWs reported being approached by the hospital management, convincing them to take the vaccine.

"*Management had to sit down and singled us out (unvaccinated), and they actually brought one of the persons who works with WHO mainly on vaccination, to talk to us, to remove the doubts we had*". Female, 29 years, HCW, vaccinated.

For some participants, being older and having chronic health conditions were described as factors that prompted them to be vaccinated. Participants with diabetes, persons living with HIV, and those with hypertension reported having accepted to receive the vaccine because of fear of the detrimental effects of COVID-19 for unvaccinated persons with these conditions.

"*My age prompted me and the underlying diseases I have. First, I have diabetes, I have high blood pressure, and I am on antiretroviral therapy*" Female, 62 years, older person, vaccinated.

"*I am not very young. Because I realized that if the disease was to strike me, maybe, I would not have a chance to fight it much*" Male, 58 years, HCW, vaccinated.

The belief in the vaccine's potential to provide protection against deadly disease played a significant role in motivating HCWs and older individuals to actively seek vaccination.

"*Fear of death because the moment you see people struggling in the ICU [intensive care unit], you will run and take these vaccines*" Female, 29 years, HCW, vaccinated.

### Interpersonal facilitators

The influence of leaders who took the vaccine was the leading social factor mentioned by some participants. Participants reported that they gained confidence that the COVID-19 vaccines were safe after witnessing religious, cultural, and political leaders, employers (hospital administrators) and peers receive the vaccine.

"*As you see the leadership going for the same (vaccination), you get motivated, like when for example, the minister of health took the jab*" Male, 58 years, HCW, vaccinated.

Some hospital administrators reported taking the vaccine to encourage their peers and the people they led to receive it.

"*There was no way that I could convince my people (juniors) to go for vaccination, yet I had not done the same*" Female, 63 years, HCW, vaccinated.

Other HCWs reported that they went in for vaccination after interacting with their peers who had taken the vaccine and had not suffered any consequences.

### Institutional factors

The risky working environment at the hospitals was a facilitator for vaccine uptake for the HCWs. Some HCWs revealed how reluctant they had been to be vaccinated until they were assigned to work directly in COVID-19 patients' wards.

"*I am always exposed. I am always among patients, so I could not think that I could never get the disease. So, I could not take it for granted, because we are always exposed among these patients*" Female, 54 years, HCW, vaccinated.

"*It is very funny. At first, I was a bit hesitant to take the vaccine. There came a time when hospital administration was looking for staff who are going to perform COVID-19 tests, and I was among the staff selected to perform COVID-19 testing. I started thinking of my safety*" Male, 35 years, HCW, vaccinated.

The implementation of mandatory COVID-19 vaccination requirements for HCWs served as a deciding factor for some HCWs, leading them to opt for vaccination to maintain their employment. Some older people reported taking the vaccine in anticipation of future mandatory movement permits in the form of COVID-19 vaccination cards.

"*So, when management sat, they decided to change the policy, that please, if you are not vaccinated, go home and get vaccinated or if you fall sick, you pay your own bills. It was hard at first until one of the persons; one staff got sick who was not vaccinated*" Male, 31 years, HCW, vaccinated.

"*At first, it was optional, we did not put much stress on everyone getting vaccinated, but as time went on, we made it mandatory*" Female, 63 years, HCW, vaccinated.

"*Even us, the health care workers, currently vaccination is a policy for the institution*" Female, 41 years, HCW, vaccinated.

### Participants' recommendations to reduce barriers to COVID-19 vaccine uptake

Participants' had several recommendations for reducing barriers to COVID-19 vaccine uptake. HCW and older people suggested using various communication channels e.g., radio, social media, posters, and megaphones to share information on COVID-19 vaccines (e.g., benefits, safety, and development processes) widely in all the communities, ensuring vaccine availability, conducting targeted vaccination outreach, putting in place mandatory vaccine policies, initiating in-country (local) manufacturing of vaccines, and financial allowances to HCWs involved in COVID-19 vaccination campaigns.

"*They should continue educating people and creating awareness. There is no other approach apart from that because if a person understands, he/she will know what exactly is taking place*" Male, 76 years, older person, not vaccinated.

"*The information should continue running on air, in communities, health workers should start going to the communities and do health education in the communities. It should be regular*" Female, 54 years, HCW, vaccinated.

Some HCWs underscored the relevance of providing enough information for people to appreciate the benefits of receiving the vaccine and cautioned that mandatory vaccination would instead force people to get fake vaccination certificates without necessarily receiving the vaccine.

"*The best thing is to counsel somebody to know why he is getting the vaccine. Yes, that is the best way because the policy will be there, and somebody will go to Nasser Road (a street in Kampala that is notorious for being a source of forged documents) and get a fake card*" Female, 44 years, HCW, vaccinated.

The HCWs and older people recommended that a regular and consistent vaccine supply to the vaccination centres would be important.

"*They have to increase the quantity of the vaccines. At least this time, people want the vaccines, but they are not available. So, they should make them available*" Male, 46 years, HCW, vaccinated.

Targeted vaccination outreach campaigns that take the vaccines were recommended as a facilitator of vaccination uptake.

"*The government should bring the vaccines and try to bring it close to people in the communities*" Male, 71 years, older person, vaccinated.

"*They should put vaccine outreaches such that people can reach every person far deep down on the ground as they have done for HIV*" Male, 38 years, HCW, vaccinated.

The HCWs and older people recommended local vaccine manufacture.

"*Uganda should work on manufacturing [its] own vaccine. [Make] Our own vaccine that is manufactured in Uganda, not from outside, as people fear vaccines from outside, which they say [are feared because] `the whites want to kill them*" Male, 21 years, HCW, vaccinated.

The recommendation for local vaccine manufacture is believed to enhance vaccine availability, supply reliability, and foster confidence in vaccine safety.

## Discussion

Our study highlights several individual, interpersonal, institutional, community, and public policy level barriers and facilitators of COVID-19 vaccine uptake among HCWs and older persons in the early phase of vaccination roll-out in Uganda in 2021.

We found out that having chronic illnesses was a barrier and at the same time a facilitator to vaccine uptake, while fear of vaccine side effects, myths and misconceptions were reported as interpersonal barriers to vaccine uptake. At the institutional level, we found out that vaccine stockouts and congestion at the vaccination centres were major health facility operational barriers. High transports costs, long distances to the vaccination centres and mobility challenges were community barriers while the prerequisite of signing consent forms before vaccination was a public policy related barrier.

Access to the correct information and older age were reported as positive intrapersonal facilitators while fear of side effects, and the influence of leaders were reported to be community facilitators for vaccine uptake. The risky working environment (hospitals) and the hospital vaccination requirements were reported as institutional requirements that promoted vaccine update.

Myths, concerns, and conspiracy theories about COVID-19 vaccines e.g., causing infertility, impotence, and weakening the body's immunity were major interpersonal barriers to vaccine uptake. Myths and conspiracy theories are well-recognized drivers of vaccine hesitancy globally [22–25]. As exposure to reliable information is a crucial intrapersonal tool for assisting people in making informed judgments [26–28], efforts to actively identify and address COVID-19 misinformation are needed to increase the acceptance of COVID-19 vaccines and other future mass vaccination campaigns, particularly those involving new vaccines.

Older age, presence of chronic health conditions, and working in a hospital setting were facilitators for the uptake of COVID-19 vaccines in the current study. These findings are consistent with those of other studies [29–31] and might be expected, since these populations were prioritised for COVID-19 vaccination due to their increased risk of infection and poor COVID-19 outcomes [32–34]. However, we also found that the presence of underlying chronic conditions was the reason for deferral or non-uptake of COVID-19 vaccination for some participants because of fears that vaccination could possibly make these conditions

worse. This underscores the need for strategies to increase knowledge about the effectiveness and safety of COVID-19 vaccines in the most-at-risk populations.

At the community level, leading by example through the public vaccination of high-profile community leaders or by HCWs getting vaccinated before vaccinating other people, facilitated uptake of vaccination. Agha and colleagues [35] in their study of COVID-19 vaccination in Nigeria showed that individuals can easily accept or refuse health campaigns depending on the attitude and perception of influential persons in their communities.

At the structural level, our study identified several barriers to COVID-19 vaccination uptake that particularly impacted older persons (non-HCWs). These included long distances to vaccination centres and associated transport costs, crowding at the health facilities, and limited vaccine supplies and/stockouts. Similar barriers were reported in other African settings during the early phase of the epidemic [36]. These barriers may have been attributed, at least in part, to limited vaccine supplies, funding shortfalls, a lack of vaccinators, sub-optimal training, inadequate planning, and COVID-19 related disruptions to essential health services [37].

Our results show both support for and caution against mandatory COVID-19 vaccination. Whereas some individuals may be encouraged to get vaccinated to stay in employment, be able to access certain public spaces and/or travel, this approach may have limitations. For example, persons who are strongly opposed to or who are not convinced of the vaccination benefits might opt to use fake COVID-19 certification. Mandatory COVID-19 vaccination could also have ethical implications if people feel forced to take something they do not wish to have which could result in the breakdown of trust between employees and their institutions [38].

In the early phase of COVID-19 vaccine roll-out in Uganda, individuals intending to receive the COVID-19 vaccines were required to sign a consent form [39]. Our results show that this requirement may have negatively impacted uptake of COVID-19 vaccination. Formal consent is not a requirement for other vaccinations in the country. COVID-19 vaccines were first made available under emergency use authorization i.e., a regulatory mechanism to facilitate the availability of unapproved product medical products, including drugs and vaccines, during a public health emergency, which required consent to be sought. The unusual requirement to consent for vaccination coupled with the misinformation about COVID-19 vaccines discussed above, may have heightened concerns about the safety of the vaccines. This finding is in agreement with the study of Bîrsanu and colleagues [40] that indicated that the unusual informed consent requirement reduced confidence in the vaccine.

## Study strengths and limitations

We enrolled individuals prioritised for COVID-19 vaccination during the early phase of COVID-19 vaccination campaign and during the second wave of COVID-19 in Uganda (May to October 2021). Thus, study participants' responses were informed by real-life experiences of the vaccination campaign including receipt of information about seeking and receiving the vaccines.

A notable limitation of this study pertains to the relatively low number older persons included in the study sample compared to the HCWs. Out of a total of 33 interviews conducted, only eight involved older people and 25 for HCWs. Furthermore, older persons interviews were exclusively sourced from a single district, Wakiso district, a mostly semi-urban/urban setting, the study results may not comprehensively reflect the lived experiences of rural older persons in relation to COVID-19 vaccine uptake. Future research should strive to include a more extensive and varied older persons participant group to enhance the comprehensiveness of the results.

## Conclusion

To maximize the success of mass vaccination campaigns it is crucial to implement a comprehensive information dissemination strategy that effectively communicates information about the vaccines. Ensuring improved access to vaccines through community outreach campaigns, maintaining adequate vaccine stocks, and providing robust health education to address vaccine misinformation and myths, may enhance COVID-19 vaccine uptake, particularly in countries where uptake remains low. Moreover, implementing these strategies can potentially increase the uptake of future vaccines during disease outbreaks.

## Supporting information

**S1 Text. In-depth interview guide for HCWs and older persons.**
(DOCX)

**S2 Text. CO-ROLL codebook.**
(DOCX)

**S3 Text. Data file.**
(DOCX)

## Acknowledgments

We would like to express our appreciation to the staff of Villa Maria Hospital, Our Lady of Consolata, Kisubi Hospital, and Entebbe Regional Referral Hospital who participated in this study as well as the entire study team at MRC/UVRI & LSHTM Uganda Research Unit.

## Author Contributions

**Conceptualization:** Sande Slivesteri, Agnes Ssali, Ubaldo M. Bahemuka, Denis Nsubuga, Moses Muwanga, Chris Nsereko, Edward Ssemwanga, Asaba Robert, Janet Seeley, Eugene Ruzagira.

**Data curation:** Sande Slivesteri, Agnes Ssali.

**Formal analysis:** Sande Slivesteri, Agnes Ssali, Denis Nsubuga, Janet Seeley.

**Funding acquisition:** Alison Elliott.

**Investigation:** Sande Slivesteri, Agnes Ssali, Janet Seeley, Eugene Ruzagira.

**Methodology:** Agnes Ssali, Janet Seeley, Alison Elliott, Eugene Ruzagira.

**Project administration:** Agnes Ssali, Ubaldo M. Bahemuka, Moses Muwanga, Chris Nsereko, Edward Ssemwanga, Asaba Robert, Alison Elliott, Eugene Ruzagira.

**Resources:** Agnes Ssali, Alison Elliott, Eugene Ruzagira.

**Supervision:** Agnes Ssali, Ubaldo M. Bahemuka, Moses Muwanga, Chris Nsereko, Edward Ssemwanga, Asaba Robert, Alison Elliott, Eugene Ruzagira.

**Validation:** Sande Slivesteri, Agnes Ssali, Denis Nsubuga, Janet Seeley.

**Visualization:** Sande Slivesteri, Agnes Ssali, Ubaldo M. Bahemuka, Denis Nsubuga, Janet Seeley, Alison Elliott, Eugene Ruzagira.

**Writing – original draft:** Sande Slivesteri, Agnes Ssali, Denis Nsubuga, Eugene Ruzagira.

**Writing – review & editing:** Sande Slivesteri, Agnes Ssali, Ubaldo M. Bahemuka, Janet Seeley, Alison Elliott, Eugene Ruzagira.

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
