## [Decision Letter · Decision Letter 0]

25 Sep 2023

PGPH-D-23-01213

Structural, Social, and Contextual factors influencing COVID-19 vaccine uptake: A qualitative methods study among Healthcare Workers and Older People in Uganda.

Dear Dr. Slivesteri,

Thank you for submitting your manuscript to PLOS Global Public Health. After careful consideration, we feel that it has merit but does not fully meet PLOS Global Public Health’s publication criteria as it currently stands. Therefore, we invite you to submit a revised version of the manuscript that addresses the points raised during the review process.

Please note that we have only been able to secure a single reviewer to assess your manuscript. We are issuing a decision on your manuscript at this point to prevent further delays in the evaluation of your manuscript. Please be aware that the editor who handles your revised manuscript might find it necessary to invite additional reviewers to assess this work once the revised manuscript is submitted. However, we will aim to proceed on the basis of this single review if possible. 

The manuscript has been evaluated by a single reviewer, and their comments are available below. Reviewer #1 has concerns regarding the number of interviewed participants, and recommend recruiting additional elderly people or alternatively, focusing your manuscript on health care workers. They also recommend additional clarity in the Methods section as it relates to how the sample number was devised and how analysis of results were performed. Could you please revise the manuscript to carefully address the concerns raised? 

We look forward to receiving your revised manuscript.

Kind regards,

Richard Ali, PhD

Staff Editor

Journal Requirements:

Additional Editor Comments (if provided):

Reviewers' comments:

Reviewer's Responses to Questions

**Comments to the Author**

1. Does this manuscript meet PLOS Global Public Health’s publication criteria? Is the manuscript technically sound, and do the data support the conclusions? The manuscript must describe methodologically and ethically rigorous research with conclusions that are appropriately drawn based on the data presented.

Reviewer #1: Partly

2. Has the statistical analysis been performed appropriately and rigorously?

Reviewer #1: N/A

3. Have the authors made all data underlying the findings in their manuscript fully available (please refer to the Data Availability Statement at the start of the manuscript PDF file)?

Reviewer #1: Yes

4. Is the manuscript presented in an intelligible fashion and written in standard English?

Reviewer #1: Yes

5. Review Comments to the Author

Reviewer #1: Thank you for the opportunity to review this interesting and important work. This is a qualitative, interview-based study of COVID-19 vaccination barriers and facilitators among Health care workers and the elderly in Uganda. The topics addressed are extremely important and the project was theoretically sound. I did have the following concerns with the manuscript, however.

1. The number of people interviewed was low. Only eight of your 33 interviews were with elderly people, and they were all from the same district. Elderly people and health care workers may not have much in common. I would suggest splitting them into two populations, which may involve recruiting more elderly people. I hate to suggest to make this paper focus only on health care workers, but I am not convinced that the number and diversity of you elderly population is enough to draw some of the conclusions you draw. Taking out the elderly people and making it a paper about health care workers may focus and strengthen the manuscript.

2. Please go into greater detail in the methods section on how you selected the number of people you chose to survey

3. Please discuss more in the methods section how you performed your results coding and how many people verified the conclusions from each interview.

6. PLOS authors have the option to publish the peer review history of their article (what does this mean?). If published, this will include your full peer review and any attached files.

**Do you want your identity to be public for this peer review?** For information about this choice, including consent withdrawal, please see our Privacy Policy.

Reviewer #1: No

---

## [Decision Letter · Decision Letter 1]

21 Feb 2024

PGPH-D-23-01213R1

Structural, Social, and Contextual factors influencing COVID-19 vaccine uptake: A qualitative methods study among Healthcare Workers and Older People in Uganda.

Dear Dr. Slivesteri,

Thank you for submitting your manuscript to PLOS Global Public Health. After careful consideration, we feel that it has merit but does not fully meet PLOS Global Public Health’s publication criteria as it currently stands. Therefore, we invite you to submit a revised version of the manuscript that addresses the points raised during the review process.

The manuscript has been evaluated by two reviewers, and their comments are available below.

Although reviewer 1 is satisfied with the revisions to the manuscript, reviewer 2 has raised a number of concerns, and makes several requests aimed at improving the clarity of the manuscript and the detail of the reporting.

Could you please carefully revise the manuscript to address all comments raised?

We look forward to receiving your revised manuscript.

Kind regards,

Steve Zimmerman, PhD

PLOS Staff Editor

Journal Requirements:

Additional Editor Comments (if provided):

Reviewers' comments:

Reviewer's Responses to Questions

**Comments to the Author**

1. If the authors have adequately addressed your comments raised in a previous round of review and you feel that this manuscript is now acceptable for publication, you may indicate that here to bypass the “Comments to the Author” section, enter your conflict of interest statement in the “Confidential to Editor” section, and submit your "Accept" recommendation.

Reviewer #1: All comments have been addressed

Reviewer #2: (No Response)

2. Does this manuscript meet PLOS Global Public Health’s publication criteria? Is the manuscript technically sound, and do the data support the conclusions? The manuscript must describe methodologically and ethically rigorous research with conclusions that are appropriately drawn based on the data presented.

Reviewer #1: Yes

Reviewer #2: No

3. Has the statistical analysis been performed appropriately and rigorously?

Reviewer #1: N/A

Reviewer #2: N/A

4. Have the authors made all data underlying the findings in their manuscript fully available (please refer to the Data Availability Statement at the start of the manuscript PDF file)?

Reviewer #1: Yes

Reviewer #2: Yes

5. Is the manuscript presented in an intelligible fashion and written in standard English?

Reviewer #1: Yes

Reviewer #2: No

6. Review Comments to the Author

Reviewer #1: Thank you for addressing the concerns from the last review

Reviewer #2: In the context of establishing clear and coherent narrative of the study argument, the following observations are made and suggested revision recommended.

1. The title needs minor but very important syntax modifications. This study, in no way can measure or infer the level of “influence” going by the study design and methodological approaches used. Some of the constructs mentioned in the title for measurement, particularly contextual factors, appear vague and redundant. A more appropriate title suggested would be; “Structural and Social factors involved with COVID-19 vaccine uptake among Healthcare Workers and Older People in Uganda: A qualitative analysis”

2.Abstract: The objective of the study as stated in the abstract needs to be appropriately stated to adequately align with the problem phenomenon warranting the study which has not been well stated. Methodology needs editorial revising to establish clarity.

3. Introduction/Background of the study: Structural, Social, and Contextual factors influencing COVID-19 vaccine uptake: A

qualitative methods study among Healthcare Workers and Older People in Uganda.

The background has not operationalize the principles extant in the title and objective of the study. The coherence of the thesis of the study is weak and has not established any argument needing to demonstrate proof of concept warranting data collection and analysis. The statement in Lines 56 - 59 is redundant and has not contributed to the context of the narrative.

There are some grammatical issues to be treated. Line 128. Line 55: “had caused 12.2 million confirmed cases (256,542 deaths) in Africa and 170,544 confirmed cases (3,632 deaths) in Uganda…” should read; “...had caused 12.2 million confirmed cases with 256,542 deaths in Africa and 170,544 confirmed cases with 3,632 deaths in Uganda…”.

Fallout from an analysis of the background narrative:

a. What is the problem and argument the study seeks to establish? This is not succinctly apparent in the background narrative. At least, from lines 54-69 should have appropriately introduced the basis of delayed vaccine rollout, efficacy expected and likely problems to be encountered with such modality, considering global antecedents with vaccine acceptance challenges with other vaccines already in use for different communicable diseases before providing citations of study demonstrating acceptance challenges with the vaccines for Covid-19 as in lines 64, 70-79. Introducing the statement “Structural, social, and contextual factors may contribute to vaccine confidence and affect the uptake of the COVID-19 vaccine among HCWs and older people in Uganda” in line 80 has not contributed appropriately to coherence and clarity of the study argument. What is meant by structural factors, or social factors or contextual factors?

b. By stating “We set out to explore the structural, social, and contextual factors that influence the uptake of COVID-19 vaccines among HCWs and older people aged 50 years using the Social Ecological Model (SEM) to structure our analysis” introduces a confusion of whether a statement of objective is intended or introduction of the theoretical framework.

c. The theoretical framework: Theories and models are very important tools in public health and health promotion research because they offer the basis for understanding individual behaviour and those factors that are associated with such behaviour. McLeroy et al (1987) clarified the importance of models within the context of an ecological perspective stated that behaviour results from the interaction between the individual and the environmental determinants, such as biological, psychological, socio-cultural and structural spheres which facilitates the behavioural outcomes. The choice of the Social Ecological Model (SEM) in this study appears appropriate. However, it must be noted that it is a framework that provides only context of levels of relationships and interactions but does not provide constructs to be measured and therefore, requires further analysis of the dynamics involved at the various levels to appropriately apply SEM to operationalize the dynamics involved at each level of the context. Any application of SEM without recourse to this micro-level consideration would easily be deemed frivolous and flawed. In the statement of lines 93-95; “Using this framework, we explored participants' knowledge, beliefs, and attitudes about COVID-19, personal experience about COVID-19 vaccines, facilitators and barriers to vaccine uptake, and opinions on the future of vaccine rollout” raises the questions of on what scientific basis has the constructs of knowledge, beliefs, and attitudes about COVID-19 been derived? Kindly review the theoretical framework and re-define the applications of SEM to operational sources of the constructs at the intra-personal, inter-personal, community and policy levels. You may consider the health belief model at the intra-, social learning at the inter- and probably social marketing or adoption of innovation at the community-, and policy levels. There are variables/constructs fitting these specified models to adequately align the thesis of the study with regards to figure 2 representations and the methodology.

4. The results: The exclusively qualitative approach has constituted a weakness in the appropriate application of the framework to adequately answer the emerging research questions in the study. I would have thought that the qualitative aspect of the study design to address context-relevance and quantitatively establish the proof of associations between structural, social and the outcome variable of covid-19 vaccine uptake.

7. PLOS authors have the option to publish the peer review history of their article (what does this mean?). If published, this will include your full peer review and any attached files.

**Do you want your identity to be public for this peer review?** For information about this choice, including consent withdrawal, please see our Privacy Policy.

Reviewer #1: No

Reviewer #2: **Yes: **Nnodimele Onuigbo ATULOMAH

---

## [Decision Letter · Decision Letter 2]

30 Apr 2024

Structural and Social factors affecting COVID-19 vaccine uptake among Healthcare Workers and Older People in Uganda: A qualitative analysis

PGPH-D-23-01213R2

Dear MR. Slivesteri,

We are pleased to inform you that your manuscript 'Structural and Social factors affecting COVID-19 vaccine uptake among Healthcare Workers and Older People in Uganda: A qualitative analysis' has been provisionally accepted for publication in PLOS Global Public Health.

Please note that the reviews you see below are what we suspect were from an earlier version of the manuscript. We tried to engage the reviewer to update with the most recent draft but they were unresponsive. There is no need to submit a response to the reviews below - this paper will now move along to production.

Best regards,

Julia Robinson

Executive Editor

Reviewer Comments (if any, and for reference):

Reviewer's Responses to Questions

**Comments to the Author**

1. If the authors have adequately addressed your comments raised in a previous round of review and you feel that this manuscript is now acceptable for publication, you may indicate that here to bypass the “Comments to the Author” section, enter your conflict of interest statement in the “Confidential to Editor” section, and submit your "Accept" recommendation.

Reviewer #2: (No Response)

2. Does this manuscript meet PLOS Global Public Health’s publication criteria? Is the manuscript technically sound, and do the data support the conclusions? The manuscript must describe methodologically and ethically rigorous research with conclusions that are appropriately drawn based on the data presented.

Reviewer #2: No

3. Has the statistical analysis been performed appropriately and rigorously?

Reviewer #2: Yes

4. Have the authors made all data underlying the findings in their manuscript fully available (please refer to the Data Availability Statement at the start of the manuscript PDF file)?

Reviewer #2: Yes

5. Is the manuscript presented in an intelligible fashion and written in standard English?

Reviewer #2: No

6. Review Comments to the Author

Reviewer #2: Response of authors to suggestions for title modification: In the second round of review, a suggestion to improve the rendering of the title was made for syntax and technical reasons to “Structural and Social factors involved with COVID-19 vaccine uptake among Healthcare Workers and Older People in Uganda: A qualitative analysis”, but no change has been made. The syntax error and technical inaccuracy of the validity of the implication of the word “influence” used in the title still remain.

Response of authors to suggestions for abstract modification: In the second round of review of the manuscript, it was pointed out that the abstract needed to be appropriately stated, particularly the background and objective of the study, to adequately align with the problem phenomenon warranting the study which has not been well stated; this observed inadequacy still remain unattended to. Stating the study objective as;“ To inform the vaccine rollout efforts, we set out to explore the social and structural factors that influenced the uptake of COVID-19 vaccines among HCWs and older people in Uganda.” is inappropriate, weak and unattainable. In the statement “To inform the vaccine rollout efforts,…” has not communicated anything. To inform the vaccine rollout effort about what? A suggested removal of the word “influence” for an observational study for which this study can not demonstrate except by an experimental study design was ignored. A fitting suggestion for the statement of objective would be; “Considering the likely challenge of vaccine acceptance among the at-risk population, the study sought to explore social and structural factors likely to be associated with the uptake of COVID-19 vaccines among HCWs and older people in Uganda.”

Response of authors to suggestions for revision in the background to the study: The authors are encouraged to keep in mind the context of the challenge of the vaccine rollout being whether the vaccines would be accepted considering fear of effectiveness and safety in the emerging misinformation arousing general doubts and concerns in the population.

Some of the grammatical issues pointed out still remain in the current revised version.

7. PLOS authors have the option to publish the peer review history of their article (what does this mean?). If published, this will include your full peer review and any attached files.

**Do you want your identity to be public for this peer review?** For information about this choice, including consent withdrawal, please see our Privacy Policy.

Reviewer #2: **Yes: **Nnodimele Onuigbo ATULOMAH
